# Efficient Conditional Generation on Scale-Based Visual Autoregressive Models

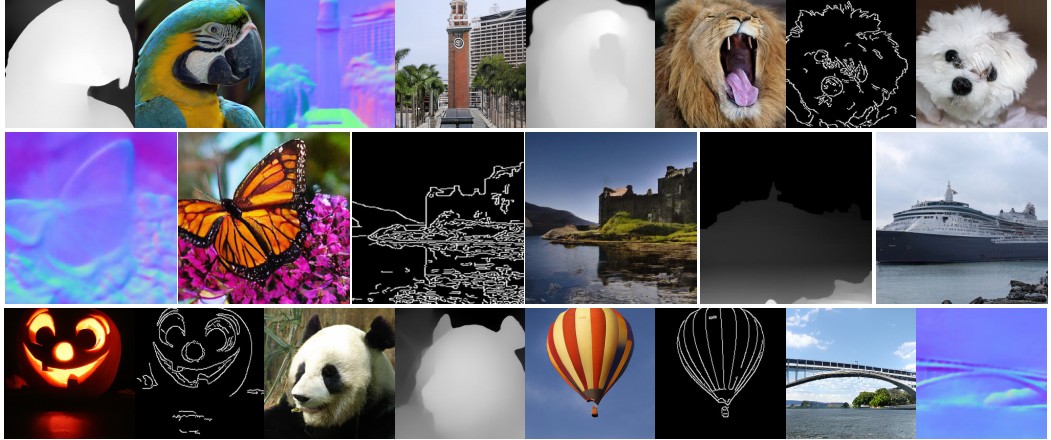

Figure 1: Visualization of ECM's conditional generation. We leverage only a 300M-parameter model to achieve high-quality conditional image synthesis at 256×256 resolution.

## Abstract

Recent advances in autoregressive (AR) models have demonstrated their potential to rival diffusion models in image synthesis. However, for complex spatially-conditioned generation, current AR approaches rely on fine-tuning the pre-trained model, leading to significant training costs. In this paper, we propose the Efficient Control Model (ECM), a plug-and-play framework featuring a lightweight control module that introduces control signals via a distributed architecture. This architecture consists of context-aware attention layers that refine conditional features using real-time generated tokens, and a shared gated feed-forward network (FFN) designed to maximize the utilization of its limited capacity and ensure coherent control feature learning. Furthermore, recognizing the critical role of early-stage generation in determining semantic structure, we introduce an early-centric sampling strategy that prioritizes learning early control sequences. This approach reduces computational cost by lowering the number of training tokens per iteration, while a complementary temperature scheduling during inference compensates for the resulting insufficient training of late-stage tokens. Extensive experiments on scale-based AR models validate that our method achieves high-fidelity and diverse control over image generation, surpassing existing baselines while significantly improving both training and inference efficiency.

## 1 Introduction

Emerging AR models such as VAR (Tian et al., 2025), LlamaGen (Sun et al., 2024), and MAR (Li et al., 2025) have shown superior performance in class- and text-to-image synthesis compared to diffusion counterparts (Peebles & Xie, 2023; Rombach et al., 2022; Podell et al., 2023; Nichol et al., 2021). However, spatial conditional generation architectures like ControlNet (Zhang et al., 2023) and T2I-Adapter (Mou et al., 2024), which provide an effective paradigm for diffusion models,

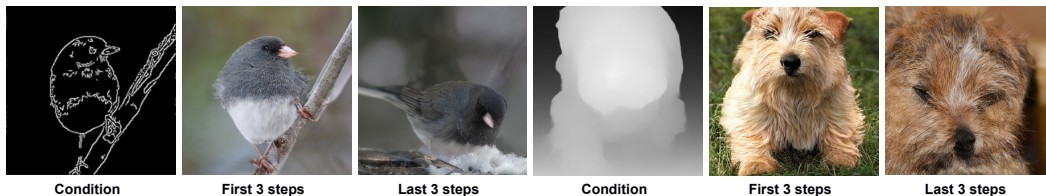

| Condition | First 3 steps | Last 3 steps | Condition | First 3 steps | Last 3 steps |

Figure 2: VAR performs 10-step AR generation for 256×256 resolution images. We inject control signals during the first and final three steps. Empirical results show that early control injection effectively guides the generation process, while late injection confers minimal control effect and risks compromising output quality.

remain less compatible for AR frameworks due to inherent differences in generative mechanisms. Specifically, AR's sequential next-token prediction versus diffusion's iterative noise refinement, hindering direct adaptation of such spatial control strategies.

Current approaches for spatial control in AR models (Li et al., 2024b;a) typically rely on fine-tuning pre-trained models with conditional inputs. Although effective, this method introduces high computational overhead, a problem that is exacerbated as generative models continue to scale. While CAR (Yao et al., 2024) adopts a ControlNet-style plug-and-play design, it requires fully training a control model that is more than half the size of the base model. Consequently, developing an efficient and effective method for conditional generation in AR models, one that adapts to their sequential token-prediction paradigm and scales with model size, has become a crucial topic to explore.

In this work, we propose a parameter-efficient, flexible, and effective plug-and-play framework, ECM, for conditional generation with scale-based AR models (Tian et al., 2025). Our method introduces a lightweight, distributed control architecture that integrates adapter layers evenly throughout the base model. This design ensures the adapters receive evolving, real-time feedback from the generation process, maintaining broad coverage to consistently steer the output. By doing so, the primary feature refining duties are shifted back to the powerful pre-trained model, allowing our control adapters to focus solely on fusing control signals at specific layers. This distributed strategy offers a more efficient alternative to other plug-and-play styles, such as ControlNet-based architectures, which typically require large, centralized modules to manage feature refinement from a fixed initial input. To further enhance our model's efficiency and coherence, we employ a partial layer sharing mechanism. By sharing the feed-forward network across adapters while leaving their attention modules independent, and pairing this with a position-aware gating mechanism, we encourage the joint learning of control features that can transition smoothly between neighboring adapters.

As shown in Figure 2, early-stage control in scale-based AR models is more effective than late-stage control, a phenomenon also observed in diffusion models (Mou et al., 2024; Balaji et al., 2022). To capitalize on this, we selectively truncate training sequences to prioritize early tokens. This strategy efficiently biases the control model toward establishing foundational structural guidance during the critical early stages of generation, and due to the scale-based model's low-to-high resolution tokenization, it also significantly reduces the number of required training tokens. A drawback of this approach is that the generator exhibits reduced confidence when sampling later-stage tokens. We compensate for this with a simple temperature scheduling: by gradually reducing the sampling temperature during inference, we leverage our specialized model's confident predictions for early tokens while amplifying the probability of more confident late-stage tokens.

Quantitative experiments demonstrate that ECM outperforms existing baselines across diverse control modalities, achieving superior generation quality and diversity with a compact control model that operates without fine-tuning the pre-trained model. This parameter-efficient, plug-and-play design not only preserves the generative capabilities of the original model but also enables substantial reductions in training costs. For example, compared to ControlVAR (Li et al., 2024a) on VARd30 (Tian et al., 2025) (a 2B-parameter pre-trained model), ECM achieves superior results on conditional generation quality with canny edge (Canny, 1986), depth (Ranftl et al., 2020), and normal map (Vasiljevic et al., 2019) despite using a base model with only 300M parameters and a 58M-parameter control model. Further, our control model trains for just 15 epochs (50% fewer than ControlVAR), with each epoch requiring only 45% of ControlVAR's training time on the same pre-trained model.

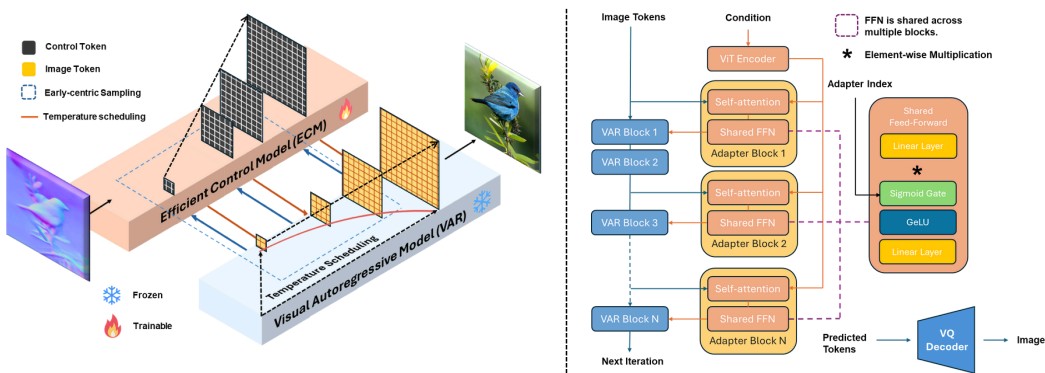

Figure 3: Workflow and architecture of ECM. On the right, the ECM architecture features multiple adapter blocks distributed evenly throughout the network. Each adapter fuses image and control tokens using element-wise addition, generating adaptive control signals. These signals are processed by a shared FFN that promotes coherent control pattern learning, while its internal layer-specific gating instills positional awareness in each adapter block. On the left, this architecture is supported by two complementary strategies: first, early-centric sampling prioritizes critical early control patterns during training for greater efficiency. Second, a temperature scheduling scheme is applied during inference, lowering the temperature for later tokens to compensate for the reduced training focus and maintain high-quality output.

## 2 RELATED WORK

### 2.1 DIFFUSION-BASED IMAGE GENERATION

Diffusion models, which synthesize images by progressively denoising Gaussian noise (Sohl-Dickstein et al., 2015), rose to prominence by surpassing Generative Adversarial Networks (Goodfellow et al., 2020) in quality and versatility. Foundational works like DDPM (Ho et al., 2020), along with numerous subsequent improvements (Song & Ermon, 2019; Nichol et al., 2021; Nichol & Dhariwal, 2021; Song & Ermon, 2020; Song et al., 2020b; 2021; Watson et al., 2021b;a; Dockhorn et al., 2021; Song et al., 2020a; Zhang & Chen, 2022; Liu et al., 2022; Karras et al., 2022; Kingma et al., 2021), rapidly advanced training and sampling efficiency. A key innovation was transitioning to latent space diffusion (Rombach et al., 2022), which enabled scalable models like the Stable Diffusion series (Rombach et al., 2022; Podell et al., 2023; Esser et al., 2024) and catalyzed applications in image editing (Hertz et al., 2022; Parmar et al., 2023; Cao et al., 2023), segmentation (Brempong et al., 2022; Xu et al., 2023), and video generation (Ho et al., 2022b; Harvey et al., 2022). More recently, architectures have shifted from UNets to Transformers, as seen in DiT (Peebles & Xie, 2023) and Stable Diffusion 3 (Yang et al., 2023), setting new state-of-the-art benchmarks.

### 2.2 AUTOREGRESSIVE-BASED IMAGE GENERATION

Autoregressive (AR) visual models, inspired by next-token prediction in NLP (Vaswani et al., 2017; Radford et al., 2019), initially operated on pixels (e.g., PixelRNN (Van Den Oord et al., 2016) with LSTM/CNN backbones (Graves & Graves, 2012; LeCun et al., 1989; He et al., 2016)) but were computationally expensive. The shift to discrete tokens, pioneered by VQ-VAE (Van Den Oord et al., 2017), improved efficiency by training AR models on quantized image representations. Recent models have built on this paradigm: LLamaGen (Sun et al., 2024) applies raster-scan prediction to these tokens, achieving performance superior to some diffusion models (Peebles & Xie, 2023; Rombach et al., 2022; Ho et al., 2022a; Dhariwal & Nichol, 2021). VAR (Tian et al., 2025), using residual quantization from RQ-VAE (Lee et al., 2022), introduces multi-scale (coarse-to-fine) tokenization for parallel sampling and state-of-the-art results. In contrast, MAR (Li et al., 2025) applies AR modeling in a continuous space using a diffusion loss. These advancements show modern AR models now rivaling diffusion models in performance, balancing trade-offs between discrete (efficiency) and continuous (quality) approaches.

### 2.3 CONDITIONAL IMAGE GENERATION

Conditional image generation, with roots in GANs (Goodfellow et al., 2020; Karras et al., 2019), advanced significantly with diffusion models. The seminal ControlNet (Zhang et al., 2023) introduced a parallel encoder to inject spatial guidance via zero-convolution layers. T2I-Adapter (Mou et al., 2024) proposed a timestep-agnostic, lightweight encoder for the same purpose. Subsequent works generalized control, with UniControl (Qin et al., 2023) using a mixture-of-experts adapter for diverse modalities and GLIGEN (Li et al., 2023b) employing gated self-attention for layout-to-image synthesis. Conditional AR generation is less explored, though methods like CAR (Yao et al., 2024) have adopted ControlNet-like architectures, with ControlAR (Li et al., 2024b) and ControlVAR (Li et al., 2024a) (see 3.2) further developing this area. Together, these innovations have enabled fine-grained control in generative models.

## 3 PRELIMINARY

### 3.1 IMAGE GENERATION WITH AUTOREGRESSIVE MODELS

Traditional visual AR models like LlamaGen (Sun et al., 2024) use a next-token prediction paradigm, decomposing images into a raster-scanned sequence of tokens $\mathbf{x} = \{x_1, x_2, ..., x_N\}$ and predicting them sequentially with a transformer parameterized by $\theta$ to maximize the joint probability:

$$p_\theta(\mathbf{x}) = \prod_{k=1}^{N} p_\theta(x_k|x_1, x_2, ..., x_{k-1}). \tag{1}$$

In contrast, scale-based AR models like VAR (Tian et al., 2025) use a next-scale prediction approach. Images are encoded across $S$ scales, where each scale $\mathbf{s}_k$ captures finer details missing from previous ones:

$$p_\theta(\mathbf{x}) = \prod_{k}^{S} p_\theta(\mathbf{s}_k|\mathbf{s}_1, \mathbf{s}_2, ..., \mathbf{s}_{k-1}), \tag{2}$$

where $\mathbf{s}_k = \{x_1, x_2, ..., x_{h \times w}\}$ is the set of tokens for a given scale. This multi-scale design allows for parallel token prediction within each scale, improving efficiency. It also creates a coarse-to-fine inductive bias—forming global structures in early scales and local details in later ones—positioning these models as a strong solution for balancing generation speed and fidelity.

### 3.2 CONDITIONAL IMAGE GENERATION WITH AUTOREGRESSIVE MODELS

We review two baseline methods for conditional AR generation. ControlAR (Li et al., 2024b), based on LlamaGen, uses a conditional-decoding strategy that fuses control tokens $\mathbf{c} = \{c_1, c_2, ..., c_N\}$ with image tokens $\mathbf{x}$ during decoding:

$$p_\theta(\mathbf{x}) = \prod_{k=1}^{N} p_\theta(x_k|\ cls + c_1, x_1 + c_2, ..., x_{k-1} + c_k), \tag{3}$$

where $cls$ is a class or start token. A key issue is that this static control signal is injected regardless of the image content, risking semantic conflicts that may require extensive fine-tuning.

In contrast, ControlVAR (Li et al., 2024a), based on VAR, employs a joint-modeling strategy that processes multi-scale image tokens $\mathbf{s}_k$ and control tokens $\mathbf{r}_k$ in parallel:

$$p_\theta(\mathbf{x}) = \prod_{k=1}^{S} p_\theta((\mathbf{s}_k, \mathbf{r}_k)|\ (\mathbf{s}_1, \mathbf{r}_1), ..., (\mathbf{s}_{k-1}, \mathbf{r}_{k-1})), \tag{4}$$

where $\mathbf{r}_k = \{c_1, c_2, ..., c_{h \times w}\}$ is the set of control tokens. While this approach effectively preserves information, its main drawback is a massive increase in the total number of tokens. For a 256x256 image, the token count can explode from 256 in a traditional AR model to 1,360 in this joint-modeling framework, risking overwhelming the model's capacity.

## 4 METHOD

In this section, we outline our proposed methodology. which consists of four key components: (1) a theoretical framework for integrating spatial control into AR models and the core architecture of our control network; (2) the gated, shared-design FFN, which optimizes the use of limited model capacity; (3) an early-centric sampling strategy, which biases the latent space to prioritize critical early-stage features and (4) temperature scheduling, which compensates for the reduced focus on late-stage learning. This combined approach refines control precision while simultaneously reducing the computational burden of token processing. A workflow of our approach is presented in Figure 3.

### 4.1 CONDITIONAL FRAMEWORK

Prior methods (Li et al., 2024b;a) for controlled generation typically learn a new joint distribution $p_\theta(\mathbf{x}|\mathbf{c})$ by conditioning a pre-trained decoder-only transformer on external control inputs, thereby modifying its original generative behavior. In contrast, our approach preserves the pre-trained distribution $p_\theta(\mathbf{x})$ by introducing a lightweight control adapter, $\mathcal{F}_\phi(.)$, parameterized by $\phi$, which dynamically steers generation by applying learned adjustments to image tokens during AR decoding. Specifically, at each generation step $k$, the adapter synthesizes context-aware control tokens $\mathcal{F}_\phi(\mathbf{r}'_k | [cls, \mathbf{s}_{<k}], \mathbf{r}_{\leq k})$, and integrate them into image tokens. Formally, we fuse control and image tokens via simple addition for both $p_\theta(.)$ and $\mathcal{F}_\phi(.)$ and formalize stepwise token alignment through a conditional decoding strategy (Li et al., 2024b):

$$p_\theta(\mathbf{x}) = \prod_{k=1}^{S} p_\theta(\mathbf{s}_k | \, cls + \mathcal{F}_\phi(1), ..., \mathbf{s}_{k-1} + \mathcal{F}_\phi(k)), \tag{5}$$

where $\mathcal{F}_\phi(k) = \mathcal{F}_\phi(\mathbf{r}'_k | [cls, \mathbf{s}_{<k}] + \mathbf{r}_{\leq k})$, the pre-trained model $p_\theta(.)$ remain frozen, while the control adapter $\mathcal{F}_\phi(.)$ is trained.

We adopt this adapter-based strategy for a crucial reason: when using a frozen pre-trained model, directly integrating control tokens risks semantic conflicts (as both encode rich semantics) and distributional misalignment (unresolvable due to the model's fixed parameters), degrading output quality. Our control adapter, through synthesizing context-aware signals by jointly processing image and control tokens, enabling adaptive fusion of task-relevant control features while preserving the semantic integrity of image tokens, alongside filtering incompatible interactions between the two distributions. By dynamically perturbing the generation trajectory through these learned adjustments, aligns decoding with spatial control without altering the base model's pre-trained distribution, thereby steering the frozen transformer toward coherent, high-quality outputs.

### 4.2 CONTROL MODEL ARCHITECTURE

Our control model's architecture is designed to inject adaptive signals into the base model. The adapter layers mirror the transformer architecture of the pre-trained model (akin to GPT-2 (Radford et al., 2019) and VQ-GAN (Esser et al., 2021)), comprising an AdaLN block for class conditioning, a self-attention layer, and an FFN. This similarity allows us to leverage pre-trained weights for faster convergence. The initial control features for these adapters are extracted by a trainable Vision Transformer (Dosovitskiy et al., 2020) whose outputs are interpolated across the scales defined by VAR (Tian et al., 2025). We intentionally omit residual connections within the adapters, as their sole purpose is to generate an additive signal for the image tokens.

### 4.3 EFFICIENT CAPACITY UTILIZATION

While isolated adapter layers excel at leveraging real-time feedback from the pre-trained models and enable flexible architectural arrangements, their hierarchical design, where subsequent adapters must compensate for residual control injection from preceding layers, risks fragmenting learned features, undermining output coherence and underutilizing limited model capacity. To address this, we propose **partial layer sharing**: self-attention blocks remain isolated to preserve their ability to model complex layer-specific spatial relations, while FFNs are shared across adapter layers. This hybrid approach establishes the shared FFN as a "common ground" to unify control signals across the network hierarchy, ensuring harmonized feature transformations, coherent generation, and efficient

use of adapter capacity. By balancing specialization (via isolated attention) and signal consolidation (via shared FFN), the design mitigates fragmentation while retaining architectural flexibility.

As illustrated in Figure 4, the attention patterns in the VAR transformer exhibit a smooth global-to-local transition as layers deepen, suggesting that distinct adapter layers specialize in handling varying attention patterns, with neighboring layers potentially sharing features modifiable via scaled adjustments. To align with this hierarchical behavior, we introduce **layer-specific gating**, where a 1D trainable parameter per adapter, modulated by a sigmoid function, is element-wise multiplied with the intermediate outputs of the shared FFN. This conditions

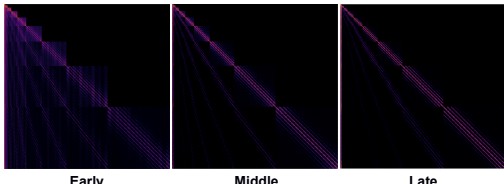

Figure 4: Analysis of attention from the early, middle, and late stages of VAR. The results reveal a transition, shifting from global structures (early stage) to localized features (late stage).

the FFN's output on layer position, enabling adaptive tuning to evolving attention patterns. The gating mechanism disentangles feature learning across distant layers, mitigates interference from unrelated depths, and promotes scalable feature learning through simple reweighting. By softly steering shared FFN capacity based on hierarchical depth, the design fosters coherent joint feature learning while introducing minimal cost in control model complexity.

### 4.4 SPECIALIZING ON EARLY TOKENS

Our experiments (Figure 2) validate that autoregressive models follow a generation pattern similar to that of diffusion models (Balaji et al., 2022; Mou et al., 2024), where early-stage control injection dominates the final result while late-stage injection imposes only minimal effects. This observation inspired us to bias the training process toward the initial part of the token sequence. We propose **early-centric sampling**, a training strategy that truncates AR sequences to prioritize early generation scales. In a VAR generation process spanning S scales, rather than training on all tokens across all $S$ scales, we sequentially sample tokens up to a dynamically chosen scale $s$ (where $s \leq S$). The likelihood of selecting scale $s$ is governed by a sampling function $\gamma(s)$ (detailed in 5.4), which we bias toward earlier scales (e.g., prioritizing $s \leq \frac{S}{2}$).

While this specialization makes our model a strong early-stage sampler, the training strategy neglects the viability of later tokens. Although these tokens are less impactful compared to early ones, insufficient training on them can still hinder overall generation quality. For an autoregressive model, this drawback can be easily compensated for during inference. Since the model is consequently less confident when sampling late-stage tokens, we employ **temperature scheduling** that gradually decreases the temperature as the generation process moves toward its later stages. Doing so guides the specialized model to discard uncertain tokens while imposing negligible computational overhead.

This synergistic approach of early-centric training and temperature scheduling significantly reduces computational cost by limiting the tokens processed per iteration, while simultaneously amplifying effectiveness by concentrating model capacity on learning critical early-stage generation patterns.

## 5 EXPERIMENT

### 5.1 EXPERIMENTAL SETUP

**Datasets.** In our experiments, we investigate class-to-image generation using the ImageNet-1k dataset (Deng et al., 2009) at 256×256 resolution. We extract canny edge (Canny, 1986), depth (Ranftl et al., 2020) and normal (Vasiljevic et al., 2019) map for testing controllability and generation quality on our control framework.

**Evaluation and metrics.** Following the standardized evaluation protocol established in prior work (Dhariwal & Nichol, 2021), we assess the performance of our model on the ImageNet validation set by reporting Fréchet Inception Distance (FID) (Heusel et al., 2017), Inception Score (IS) (Salimans et al., 2016), Precision, and Recall (Kynkäänniemi et al., 2019). Additionally, we measure F1-score for canny and RMSE for depth and normal. These metrics collectively quantify both the perceptual quality and diversity of generated images.

Table 1: Quantitative results of class-to-image (C2I) conditional generation. ECM on 310M-model has 58M parameters, 78M on 600M-model and 196M on 1.0B-model. #Prm. refers to the number of parameters of the pre-trained model. Results marked with "*" are estimated from histograms of the reported work, while † indicates we do not have access to the model. "+T.S." indicates using temperature scheduling during inference.

| Methods | #Prm. | Canny | | | | | Depth | | | | | Normal | | | | |
|---|---|---|---|---|---|---|---|---|---|---|---|---|---|---|---|---|
| | | FID↓ | IS↑ | Pre.↑ | Rec.↑ | F1↑ | FID↓ | IS↑ | Pre.↑ | Rec.↑ | RMSE↓ | FID↓ | IS↑ | Pre.↑ | Rec.↑ | RMSE↓ |
| ControlVAR* | 310M | 16.20 | 81 | 0.67 | 0.56 | | 13.80 | 95 | 0.72 | 0.51 | | 14.20 | 90 | 0.70 | 0.54 | |
| (Li et al., 2024a) | 600M | 13.00 | 94 | 0.69 | 0.56 | - | 13.40 | 97 | 0.67 | 0.51 | - | 12.90 | 98 | 0.68 | 0.53 | - |
| | 1.0B | 15.70 | 99 | 0.59 | 0.57 | | 12.50 | 123 | 0.68 | 0.44 | | 11.50 | 122 | 0.67 | 0.51 | |
| | 2.0B | 7.85 | 161 | **0.74** | 0.49 | | 6.50 | 181 | 0.77 | 0.42 | | 6.60 | 172 | **0.76** | 0.46 | |
| ControlAR† | 111M | 10.64 | | - | | 34.15 | 6.67 | | - | | 32.41 | | | - | | |
| (Li et al., 2024b) | 343M | 7.69 | | | | 34.91 | 4.19 | | | | **31.11** | | | | | |
| | 310M | 5.77 | 181 | 0.71 | 0.64 | 36.81 | 3.52 | 218 | 0.77 | 0.59 | 33.41 | 3.93 | 205 | 0.75 | 0.61 | 24.54 |
| | +T.S. | 5.27 | 189 | 0.71 | 0.64 | 37.13 | 3.37 | 225 | 0.78 | 0.59 | 33.24 | 3.85 | 207 | 0.75 | 0.61 | **24.47** |
| ECM | 600M | 5.64 | 189 | 0.70 | 0.65 | 36.42 | 3.24 | 234 | 0.76 | 0.61 | 33.98 | 3.89 | 216 | 0.73 | 0.64 | 24.81 |
| (Ours) | +T.S. | 5.13 | 196 | 0.71 | 0.65 | 37.05 | 3.17 | 235 | 0.76 | 0.61 | 33.77 | 3.72 | 221 | 0.73 | 0.64 | 24.71 |
| | 1.0B | 5.28 | 197 | 0.71 | 0.66 | 37.48 | 2.88 | 246 | 0.77 | 0.61 | 34.07 | 3.68 | 227 | 0.74 | 0.64 | 24.85 |
| | +T.S. | **5.11** | **200** | 0.71 | **0.66** | **37.65** | **2.77** | **255** | **0.77** | **0.61** | 34.03 | **3.57** | **233** | 0.74 | **0.64** | 24.83 |

**Implementation details.** The control model architecture is detailed in Section 4.2. We initialize our vision transformer (Dosovitskiy et al., 2020) control encoder with pretrained DINOv2 (Oquab et al., 2023) weights and leverage the family of VAR (Tian et al., 2025) models (depth 16, 20 and 24) for C2I experiments. The self-attention block in control model is initialized using the weights from the pre-trained self-attention block at the same level, whereas the shared FFN is initialized randomly. The gate parameter is initialized with a Gaussian distribution ($\mu = 4$, $\Sigma = 1$) to ensure it starts in a nearly fully open state. Training adheres to VAR's protocol: AdamW (Kingma & Ba, 2014) optimizer ($\beta_1 = 0.9$, $\beta_2 = 0.95$, weight decay 0.05), base learning rate is $10^{-4}$. We train our control model for 15 epochs with the batch size of 128. To enable classifier-free guidance (Ho & Salimans, 2022), class and control tokens are randomly dropped with a 10% probability. For inference, we adopt VAR's top-$k$ and top-$p$ sampling strategy, with quantitative results reported using a classifier-free guidance strength of 3.0 (other hyperparameters are detailed in Appendix A).

## 5.2 QUANTITATIVE ANALYSIS

**Class-to-image conditional generation.** We compare our method with ControlAR (Li et al., 2024b) and ControlVAR (Li et al., 2024a), two state-of-the-art baselines on LlamaGen (Sun et al., 2024) and VAR (Tian et al., 2025), which have demonstrated surpassing diffusion-based methods. We jointly train on three types of conditions: canny edge (Canny, 1986), depth (Ranftl et al., 2020), and normal map (Vasiljevic et al., 2019). Without fine-tuning, our method outperforms all baselines on VARd16 and significantly surpasses ControlVAR on VARd30, a method operating within the same model family, despite using a control model of only 58M parameters (approximately 20% of the base model). Our method exhibits strong spatial control, achieving F1-scores and RMSE values on par with the baselines. However, a crucial pattern emerges with scale: while perceptual metrics like FID (Heusel et al., 2017) and IS (Salimans et al., 2016) continue to improve on larger models, these spatial accuracy scores remain rather stable. This indicates that the cross-entropy objective successfully prioritizes closing the distributional gap, learning to generate perceptually realistic images that treat the condition as a strong guide rather than a rigid destination. The model favors plausibility over exact spatial replication. Furthermore, our method naturally handles these diverse control modalities without explicit type conditioning. More analysis is in Appendix B.

**Training and inference efficiency.** We compare our method with ControlVAR (Li et al., 2024a), a method operating on the same VAR (Tian et al., 2025), to evaluate the efficiency of our approach. As shown in Figure 5, our method only requires 45% training time per epoch compared to ControlVAR on VARd16, while also requiring only 15 total training epochs—half the 30 epochs needed by the ControlVAR. Furthermore, the model cost 0.23s per generation on single A100 GPU under FP16 with batch size equals to 1, introduces only minor inference overhead comparing with VAR's 0.19s.

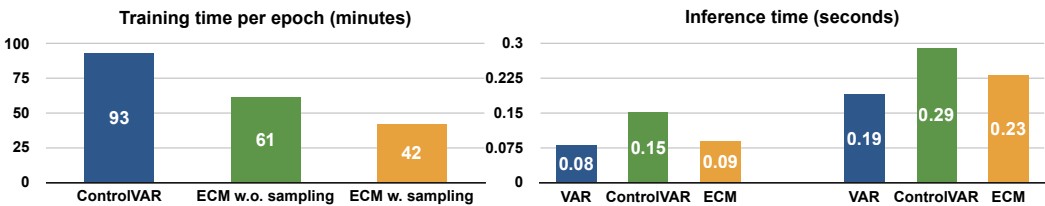

Figure 5: Training time shows that our method achieves substantial reductions in training costs (w.o. and w. sampling means without and with early-centric sampling). For inference time (per-image generation time), the left part is conducted using batch size equals 5 and the right equals 1).

Table 2: (a) Analysis of scalability with different setups of adapter layers and shared FFNs (details in Appendix A). (b) Analysis of how adapter layer positions affects model's performance.

| #Adapter | #FFN | FFN Ratio | #Param. | FID↓ | IS↑ |
|---|---|---|---|---|---|
| 3 | 1 | 4 | 52M | 5.80 | 179.07 |
| 4 | 1 | 4 | 58M | 5.77 | **181.36** |
| 5 | 1 | 4 | 64M | **5.56** | 180.79 |
| 4 | 2 | 4 | 69M | 5.85 | 180.90 |
| 4 | 1 | 6 | 62M | 5.82 | 179.02 |

(a)

| Layer position | 1, 5, 9, 13-th | 1, 6, 11, 16-th |
|---|---|---|
| FID↓ | 6.47 | **6.23** |
| IS↑ | 138.5 | **139.56** |
| Pre.↑ | **0.73** | **0.73** |
| Rec.↑ | **0.82** | **0.82** |

(b)

## 5.3 QUALITATIVE ANALYSIS

As illustrated in Figure 6, visual analysis comparing our method on VARd16 (Tian et al., 2025) with ControlVAR (Li et al., 2024a) on VARd30 demonstrates that despite utilizing a smaller pre-trained model, our approach achieves superior generation quality and enhanced spatial alignment. These results highlight the effectiveness and efficiency of our method.

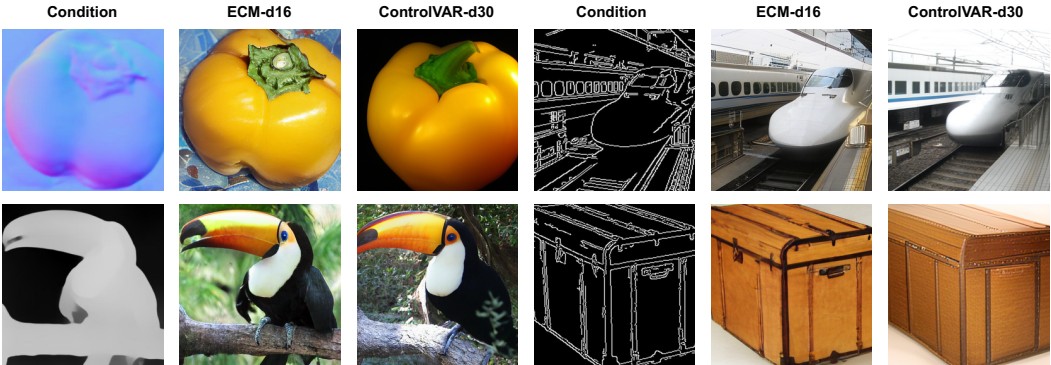

Figure 6: Comparison of conditional generation with ControlVAR (Li et al., 2024a). Our method achieves better spatial constraints while maintaining high-fidelity generation.

## 5.4 ABLATION STUDIES

We conduct ablation studies on the VAR (Tian et al., 2025) model with 16 depth, exclusively trained on canny edge (Canny, 1986). Unless specified, our control model employs four adapter layers sharing one FFN, anchored at depths 1, 6, 11, and 16.

**Scalability.** Our quantitative results demonstrate that the proposed method efficiently scales with the base model. We further explore scaling strategies on VARd16 (measured via FID (Heusel et al., 2017) and IS (Salimans et al., 2016) for canny edge (Canny, 1986)), we observe (Table 2a) that increasing the number of self-attention blocks (from 3 to 5) while retaining a single shared FFN enhances performance, likely due to expanded control coverage and smoother injection during generation. Conversely, scaling FFN (either in quantity or hidden state ratio) slightly degrades performance, potentially due to training instability from randomly initialized larger FFN blocks (unlike

Table 3: (a) Analysis indicates combining partial layer sharing and layer-specific gating, our method achieves higher performance. (b) For early-centric sampling, as the strength increases, sampling becomes more weighted toward earlier tokens. The results demonstrate both effectiveness and efficiency of this sampling strategy. $\mu(\mathbf{x})$ represents the average number of tokens in a single sequence.

| Architecture | #Params | FID↓ | IS↑ | Pre.↑ | Rec.↑ |
|---|---|---|---|---|---|
| Default | 70M | 6.23 | 139.56 | 0.73 | 0.82 |
| Default | 58M | 6.68 | 136.84 | 0.71 | 0.78 |
| + Partial Adapter Sharing | 58M | 6.39 | 138.03 | 0.72 | 0.82 |
| + Layer-specific Gating | 58M | **6.15** | **140.28** | **0.73** | **0.83** |

(a)

| Sampling Strength ($\alpha$) | $\mu(\mathbf{x})$ | FID↓ | IS↑ | Pre.↑ | Rec.↑ |
|---|---|---|---|---|---|
| No Sampling | 680 | 6.23 | 139.56 | 0.73 | 0.82 |
| 1 | 342 | **5.64** | 143.36 | 0.75 | 0.82 |
| 2 | 88 | 5.68 | 143.97 | **0.75** | **0.84** |
| 3 | 62 | 5.88 | **144.08** | **0.75** | 0.83 |
| 5 | **40** | 6.04 | 140.03 | 0.74 | **0.83** |

(b)

attention blocks which is initialized from pre-trained weights). Also, more FFNs partially block sharing between adapters, cause a shift toward local rather than global coherence learning.

**Adapter layer position.** The positioning of adapter layers critically influences model performance. To evaluate coverage effects, we tested two configurations: one placing adapters every 5 layers (at depths 1, 6, 11, and 16) and another every 4 layers (at depths 1, 5, 9, and 13). Results summarized in Table 2b demonstrate that broader spacing (5-layer intervals) yielded superior performance compared to denser placement (4-layer intervals). This insight informed our final design, where adapters are anchored at the first and final layers to ensure boundary integration, with remaining adapters evenly distributed across intermediate layers to balance comprehensive coverage.

**Partial adapter sharing and layer-specific gating.** We evaluate the impact of partial layer sharing and layer-specific gating. We start with a baseline "Default" architecture (adapters as standard transformer blocks), and observe that reducing the FFN's hidden state ratio causes significant performance degradation (Table 3a), underscoring the critical role of FFN capacity. Introducing partial sharing by consolidating all adapter FFNs into a single shared block improves metrics, as shared parameters establish a unified feature space that allows isolated adapters jointly learning coherent features. Augmenting this with layer-specific gating further boosts performance, suggesting that conditioning the shared FFN on layer position enhances feature fusion. By applying lightweight, layer-specific scalars to intermediate outputs, the design disentangles uncorrelated features while merging relevant ones, fostering coherent and hierarchical feature learning without enlarging model.

**Early-centric sampling.** We explore how different early-centric sampling strategies influence the performance of conditional generation and their impact on training efficiency. The experiment is conducted on VARd16 (Tian et al., 2025), with scales pre-defined as $(1, 2, 3, 4, 5, 6, 8, 10, 13, 16)^2$, a total of 680 tokens. Using a predefined monomial function to sample training sequences, formulated as $\gamma(s) = S(\frac{s}{S})^{\alpha}$ where $\alpha$ is the sampling strength, we demonstrate in table 3b that early-centric sampling consistently improves overall performance, validating the effectiveness of our approach. However, there is a trade-off between performance and efficiency as sampling strength varies. We observe that performance tends to converge when sampling approaches a uniform random selection, but the expected total number of tokens decreases significantly as sampling strength increases. To strike a balance between performance and efficiency, we adopt an early-centric sampling strategy with a sampling strength of 2, which optimizes both aspects effectively.

# 6 CONCLUSION

In this work, we propose Efficient Control Model (ECM), a novel framework for spatial conditional generation within scale-based AR models. Unlike traditional fine-tuning approaches, ECM employs an adapter-based design that dynamically generates control signals through real-time feedback from the pre-trained model, enabling agile and adaptive guidance without fine-tuning. Furthermore, we introduce partial adapter sharing, layer-specific gating, early-centric sampling and temperature scheduling to minimize parameter overhead and training costs while enhance generation quality. Extensive experiments demonstrate ECM's superior efficiency and effectiveness compared to baseline methods, positioning it as a competitive solution for state-of-the-art conditional generation tasks.

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

## A  IMPLEMENTATION DETAILS

**Model configurations.** For our experiments with VARd16 and VARd20 (Tian et al., 2025), we use four adapter layers with a single shared FFN and a control encoder initialized from DINOv2-Small (Oquab et al., 2023). The adapters are anchored at depths 1, 6, 11, 16 for VARd16 and 1, 7, 13, 20 for VARd20. For the larger VARd24 model, we scale to a six-adapter configuration (depths 1, 6, 11, 16, 21, 24) and use a DINOv2-Base encoder. Our ablation studies (Section 5.4) explore alternative configurations, including setups with 3 adapter layers (depths 1, 8, 16) and 5 adapter layers (depths 1, 4, 8, 12, 16). In these studies, we also evaluate a two-FFN setup, where one FFN is shared by the adapters at depths 1, 6 and a second is shared by those at 11, 16. Details results is presented in Table 4.

**Hyperparameter choices.** For our experiments, we set sampling parameters as follows: for VARd16 and VARd20 (Tian et al., 2025), we use top-$p$ = 0.96 and top-$k$ = 900, aligning with default choices in prior work. For VARd24, we maintain top-$p$ = 0.96 but reduce top-$k$ to 50, with the reasoning for this adjustment detailed in Appendix B. Our temperature scheduling follows a 2nd-order polynomial function defined as

$$T_s = T_{high} + (T_{low} - T_{high}) * (\frac{s}{S})^2 \tag{6}$$

where $T_s$ is the current temperature at scale $s \in S$. The start ($T_{high}$) and end ($T_{low}$) temperatures are set based on conditions: for canny, $T_{high}$ = 0.9 and $T_{low}$ = 0.8; for depth, $T_{high}$ = 1.0 and $T_{low}$ = 0.9 and for normal, $T_{high}$ = 1.0 and $T_{low}$ = 0.8.

**Condition image pre-processing.** During training, images are randomly cropped to 256×256 and resized to 224×224 to match the vision transformer's (Dosovitskiy et al., 2020) input dimensions. For joint training with multiple conditions, we apply NEAREST resizing to preserve sharp edges in canny maps (Canny, 1986) and BICUBIC for depth/normal (Ranftl et al., 2020; Vasiljevic et al., 2019) maps to retain gradual transitions, whereas in ablation studies (only canny is trained), BILINEAR interpolation is used to simulate general scenarios. During evaluation, the final experiments leverage OpenAI's (Dhariwal & Nichol, 2021) 10,000-image reference batch for fair comparison with prior works, while ablations employ the full 50,000 validation images to reflect realistic 1-to-1 conditional generation performance, potentially introducing discrepancies between quantitative and ablation results.

## B  ADDITIONAL ANALYSIS

**Quantitative analysis.** The performance disparity across control types arises primarily from their inherent structural differences: canny edges (Canny, 1986) impose stricter constraints by extracting sharp, detailed image features, limiting output diversity as the model adheres closely to precise contours. While depth (Ranftl et al., 2020) and normal maps (Vasiljevic et al., 2019), which outline broader object geometry, allow greater generative flexibility due to their gradual transitions. Additionally, the complexity of canny edges, encoding intricate spatial features, presents greater generalization challenges compared to the more abstract, low-frequency patterns in depth/normal representations.

**Scalability.** In addition to the patterns observed for canny edge (Section 5.2), we note a performance saturation trend across other modalities. For instance, increasing to a 5-layer adapter setup yields only marginal improvements on depth (Ranftl et al., 2020) and normal (Vasiljevic et al., 2019) maps. This suggests that without fine-tuning, performance gains plateau as adapter parameters increase, indicating that the frozen, pre-trained model itself becomes an inherent bottleneck, constraining the capacity of the control model.

We also find that as the base model scales, the vision encoder must to scale with it. Our experiments show that using a small encoder (DINOv2-Small (Oquab et al., 2023)) with a large base model (VARd24 (Tian et al., 2025)) yields marginal performance gain compared to its application on smaller models like VARd16. However, when we increase both the number of adapter layers and use a larger vision encoder (DINOv2-Base), performance continues to scale as expected. This indicates that scaling must be holistic: deploying our method on larger base models requires strengthening both the adapter architecture and the vision transformer encoder (Dosovitskiy et al.,

Table 4: Analysis of scalability with different number of adapter layers, sharing FFNs on VARd16.

| #Adapter | #FFN | FFN Ratio | #Param. | Canny | | Depth | | Normal | |
|---|---|---|---|---|---|---|---|---|---|
| | | | | FID↓ | IS↑ | FID↓ | IS↑ | FID↓ | IS↑ |
| 3 | 1 | 4 | 52M | 5.80 | 179.07 | 3.54 | 217.65 | 4.00 | 205.11 |
| 4 | 1 | 4 | 58M | 5.77 | **181.36** | 3.52 | 217.85 | **3.93** | 205.16 |
| 5 | 1 | 4 | 64M | **5.56** | 180.79 | **3.47** | 218.40 | **3.93** | 205.08 |
| 4 | 2 | 4 | 69M | 5.85 | 180.90 | 3.55 | 219.02 | 3.99 | **205.83** |
| 4 | 1 | 6 | 62M | 5.82 | 179.02 | 3.51 | **219.27** | 4.02 | 205.92 |

2020). A more powerful encoder is critical for propagating strong control signals across all depths, ensuring the distributed adapters are effectively utilized.

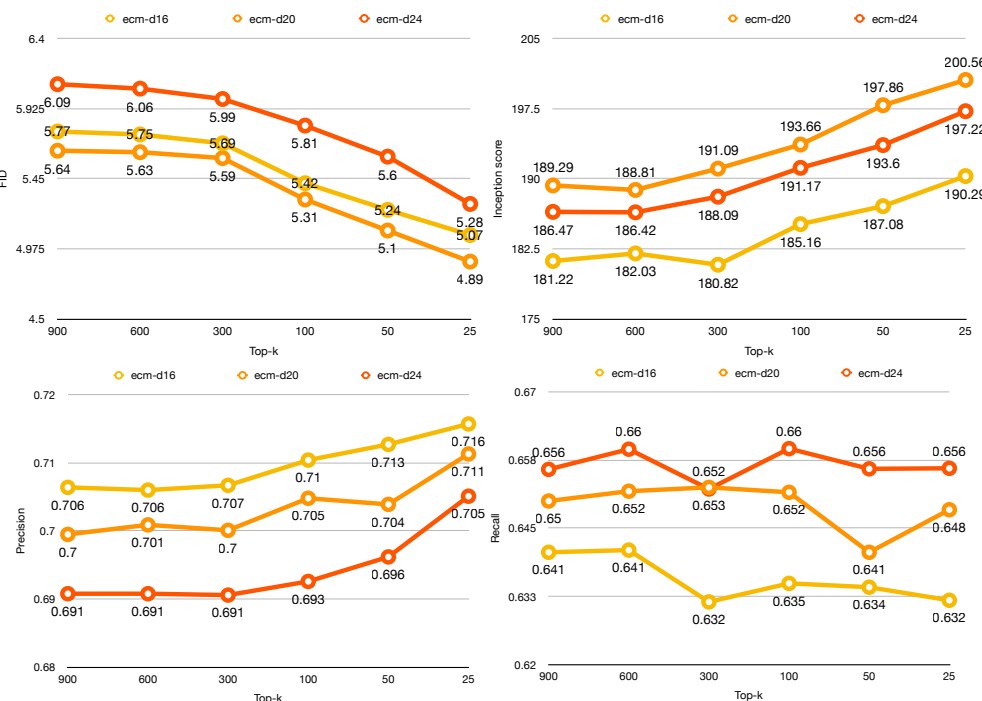

Figure 7: The effect of top-k sampling on canny edge (Canny, 1986).

Finally, we observe that a model's sampling behavior inherently changes with scale. When using identical sampling parameters (top-$k$ and top-$p$), the larger VARd24-based (Tian et al., 2025) model exhibits lower FID, IS, and Precision but higher Recall compared to its smaller counterparts (Figure 7, 8 and 9). We hypothesize this is an anticipated behavior, as larger models learn a wider combination of tokens, leading to higher-entropy output distributions. To normalize for this effect and focus on generation quality over raw creativity, we reduce the top-$k$ sampling parameter for the larger model in our reported results, while still maintaining superior Recall.

**Temperature scheduling.** Our temperature scheduling is designed to leverage the specific characteristics of our early-centric trained model. We observe that this training strategy improves the model's ability to sample high-probability tokens accurately, effectively making it a better "sampler" rather than a more creative "generator." Consequently, applying a higher initial temperature encourages more diverse, albeit potentially less accurate, early token generation. As the process moves toward the later stages, where tokens received less training, we gradually reduce the temperature. This makes the sampling more deterministic, effectively pruning uncertain tokens and enhancing the reliability of the final output.

Based on this, our typical approach starts with a temperature near the default (1.0) and decreases it over time. A notable exception is the canny edge, where a slightly lower initial temperature yielded

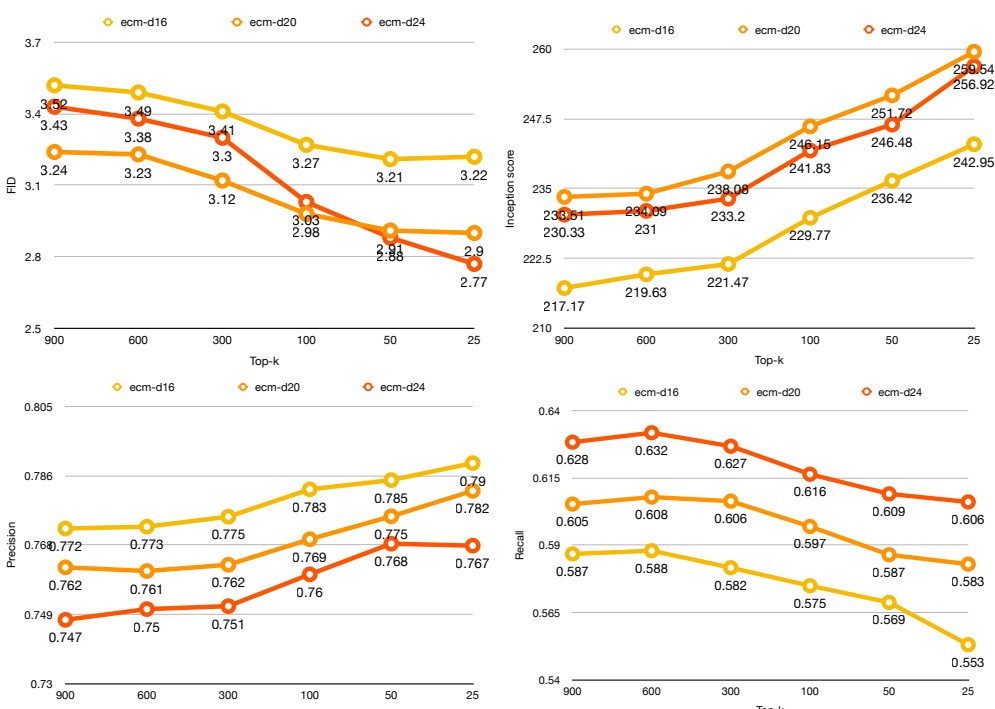

Figure 8: The effect of top-k sampling on depth map (Ranftl et al., 2020).

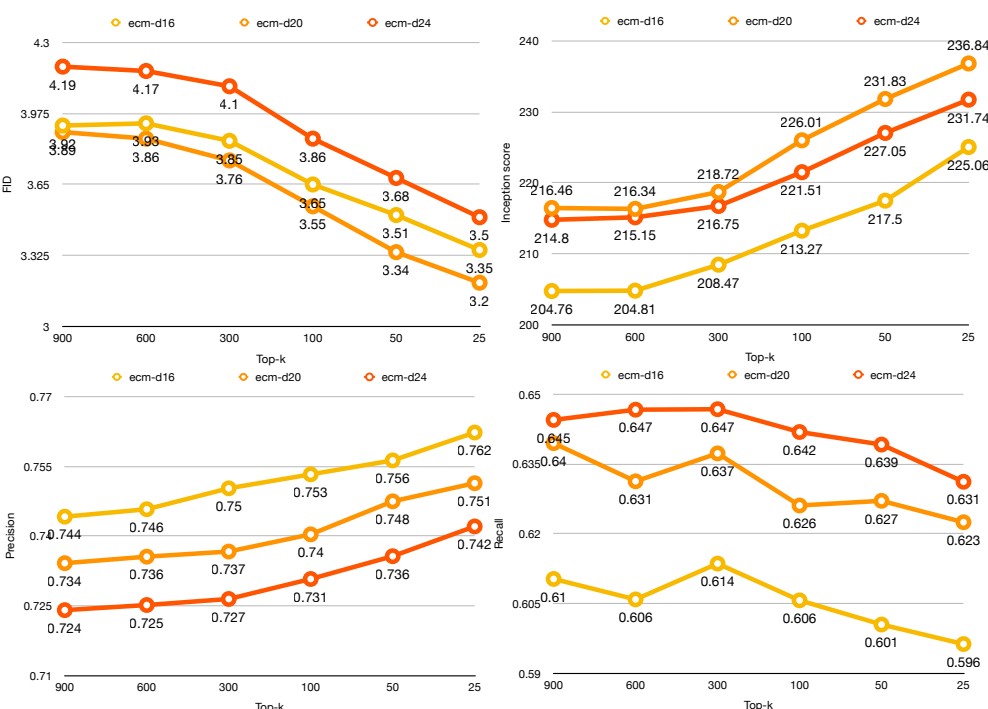

Figure 9: The effect of top-k sampling on normal map (Vasiljevic et al., 2019).

better performance. We attribute this to the higher complexity of canny edge; a more deterministic initial sampling phase is likely more beneficial for capturing their intricate structures accurately from the start.

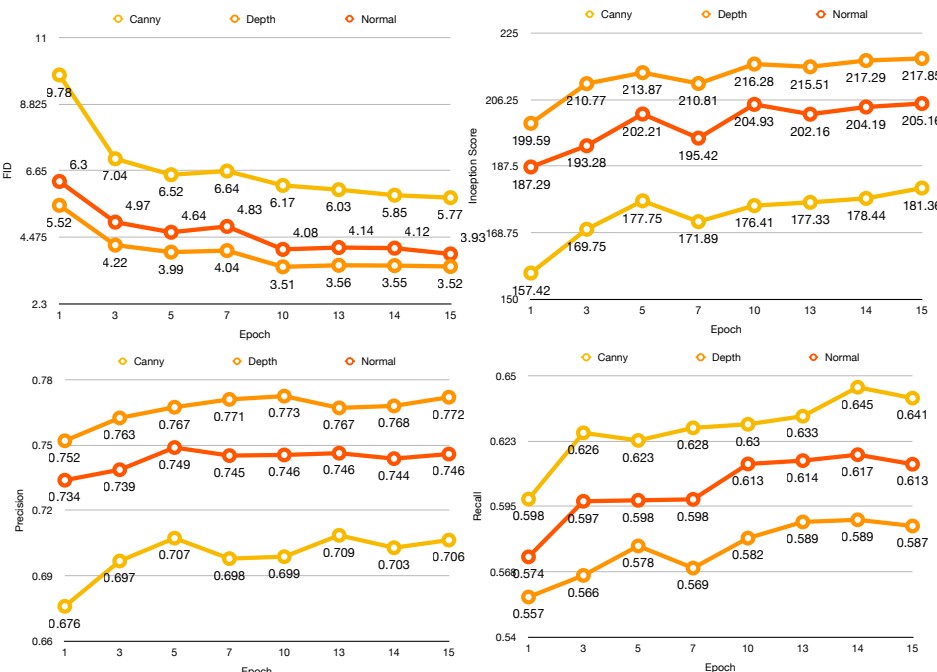

Figure 10: The training curve on VARd16 (Tian et al., 2025). Most convergence happens before 10 epochs.

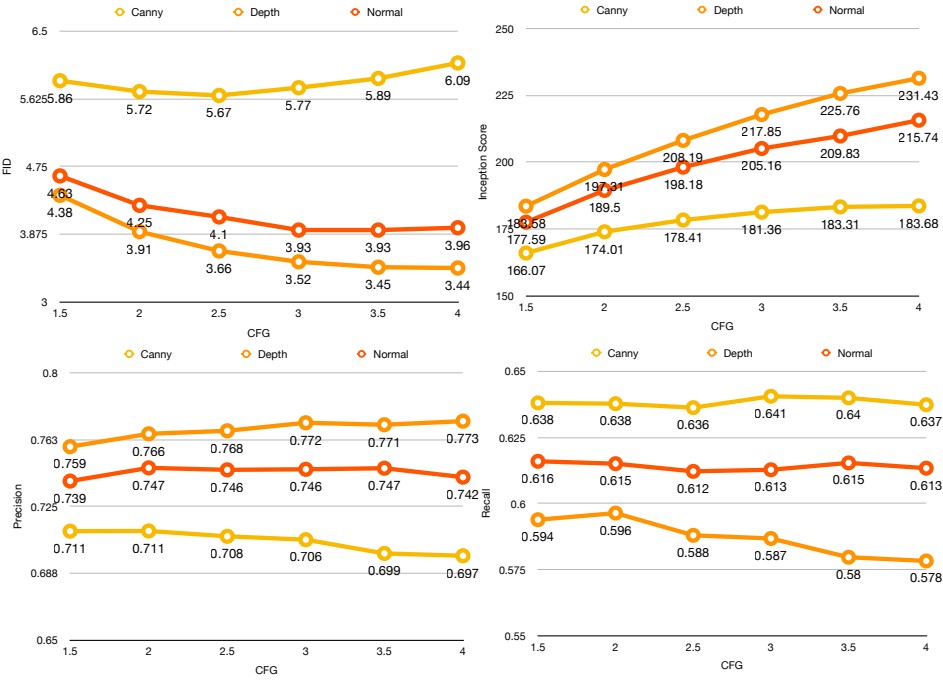

Figure 11: The effect of classifier-free guidance (Ho & Salimans, 2022). CFG exhibits varying effects across different control types.

**Training curve.** As illustrated in Figure 10, the majority of model convergence occurs within the first 10 training epochs, suggesting that the current training duration could be further shortened as a compromise between minor performance sacrifice and computational resource constraints.

**Effects of classifier-free guidance strength.** The impact of classifier-free guidance strength (CFG) (Ho & Salimans, 2022) varies across modalities. For canny edge (Canny, 1986), the optimal FID (Heusel et al., 2017) is achieved at a CFG value of 2.5 and progressively downgrade, aligning with our earlier observation that overly strong control signals may constrain generative diversity. In contrast, depth (Ranftl et al., 2020) and normal maps (Vasiljevic et al., 2019) exhibit diverging behavior: their FID scores improve progressively as CFG increases, plateauing around a value of 4. To balance these dynamics, we report quantitative results at a CFG of 3, a compromise that accommodates multiple modalities while maintaining reasonable performance as reference.

**Inference FLOPS.** The computational overhead introduced by ECM involves preprocessing control images via ViT (Dosovitskiy et al., 2020) and integrating control tokens through adapter layers, indicating the computational cost scales approximately with the number of adapter layers. We evaluated inference FLOPS on VARd16 (Tian et al., 2025) with ECM of 4 layers. For the first token prediction, measurements showed 619.05M FLOPS without ECM versus 750.78M FLOPS with ECM, while the full iterative process required 478.13G FLOPS without ECM compared to 546.92G FLOPS with ECM.

## C  TEXT-TO-IMAGE GENERATION

Since VAR lacks native text-to-image generation abilities, we adapt a variant, VAR-CLIP (Zhang et al., 2024), for spatial T2I tasks. Our experiments leverage VARd16-CLIP, trained on ImageNet1k (Deng et al., 2009) with text prompts extracted via BLIP-2 (Li et al., 2023a), using the same setup as ECM on VARd16. By retaining transformer blocks analogous to those in the pre-trained model, our approach maintains compatibility with its conditioning mechanisms. We measured FID, IS and CLIP score (Hessel et al., 2022) (in Table 5), the results affirm its potential for spatial-text-conditioned image synthesis (Visualizations are presented in Figure 12, the original quantitative results of VAR-CLIP is presented in Table 6).

Table 5: Quantitative results of text-to-image using ECM on VAR-CLIP (Zhang et al., 2024).

| Method | Canny | | | Depth | | | Normal | | |
|--------|-------|-----|-------|-------|-----|-------|--------|-----|-------|
| | FID↓ | IS↑ | CLIP↑ | FID↓ | IS↑ | CLIP↑ | FID↓ | IS↑ | CLIP↑ |
| ECM | 9.12 | 114.36 | 22.45 | 6.98 | 122.3 | 22.50 | 7.54 | 123.58 | 22.51 |

Table 6: Reported quantitative results of VAR-CLIP (Zhang et al., 2024).

| Method | FID↓ | IS↑ | CLIP↑ |
|--------|------|-----|-------|
| VAR-CLIP (Zhang et al., 2024) | 4.04 | 144 | 22.21 |

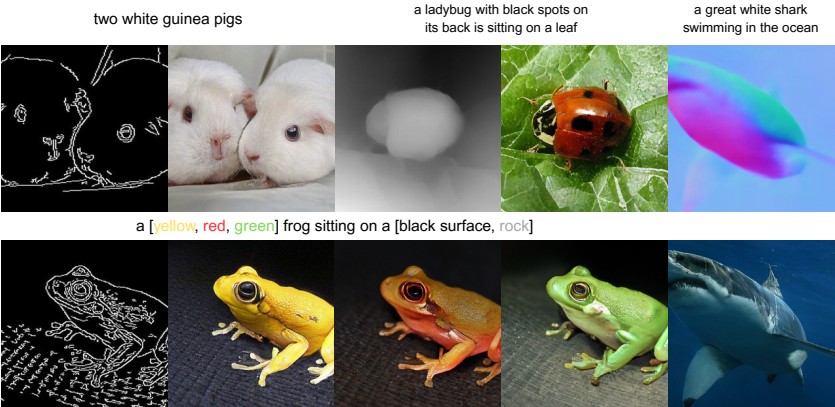

Figure 12: Visualization of Text-to-image generation using VAR-CLIPd16 (Zhang et al., 2024).

# D    LIMITATION

The discrete tokenization approach in VAR (Tian et al., 2025) introduces unexplained residuals, as noted in prior work (Tang et al., 2024), and our method's lack of explicit detail control further limits its ability to generate high-frequency features accurately. Additionally, while early-centric sampling is effectively performed on VAR, because VAR in nature preserve global awareness during the entire generation, this strategy may not generalize to AR models like LlamaGen (Sun et al., 2024), where next-token prediction inherently lacks retaining such global information throughout the generation process. Futhermore, a key disadvantage of the autoregressive paradigm is its vulnerability to noisy inputs, a trait not shared by diffusion-based methods that are natural denoisers. The results in Table 7 demonstrate this weakness: while minor noise slightly hurts performance on hard, noisy conditions like canny edges, it severely degrades generation quality on smooth conditions like depth and normal maps. Consequently, the autoregressive approach is less practical for real-world scenarios that involve imperfect or noisy data. Suggesting training with noisy samples might be necessary for deployment.

Table 7: Analysis of how noisy conditions affect performance. We inject Gaussian noise ($\mu = 0$, $\sigma = 0.1$) into control images.

| $\sigma$ | Canny | | Depth | | Normal | |
|---|---|---|---|---|---|---|
| | FID↓ | IS↑ | FID↓ | IS↑ | FID↓ | IS↑ |
| 0 | 5.77 | 181.36 | 3.52 | 217.85 | 3.93 | 205.16 |
| 0.1 | 7.61 | 155.78 | 14.75 | 115.19 | 145.58 | 10.52 |

# E    USAGE OF LLMS

During the preparation of this manuscript, we utilized large language models (LLMs) to proofread, correct grammar, and enhance the overall clarity of the text. Additionally, we used these models to help identify relevant literature and compile a comprehensive "Related Works" section.

## F  MORE VISUALIZATION

We provide more visualizations on VARd16 (Tian et al., 2025) at 256×256 resolution across multiple types of conditions.

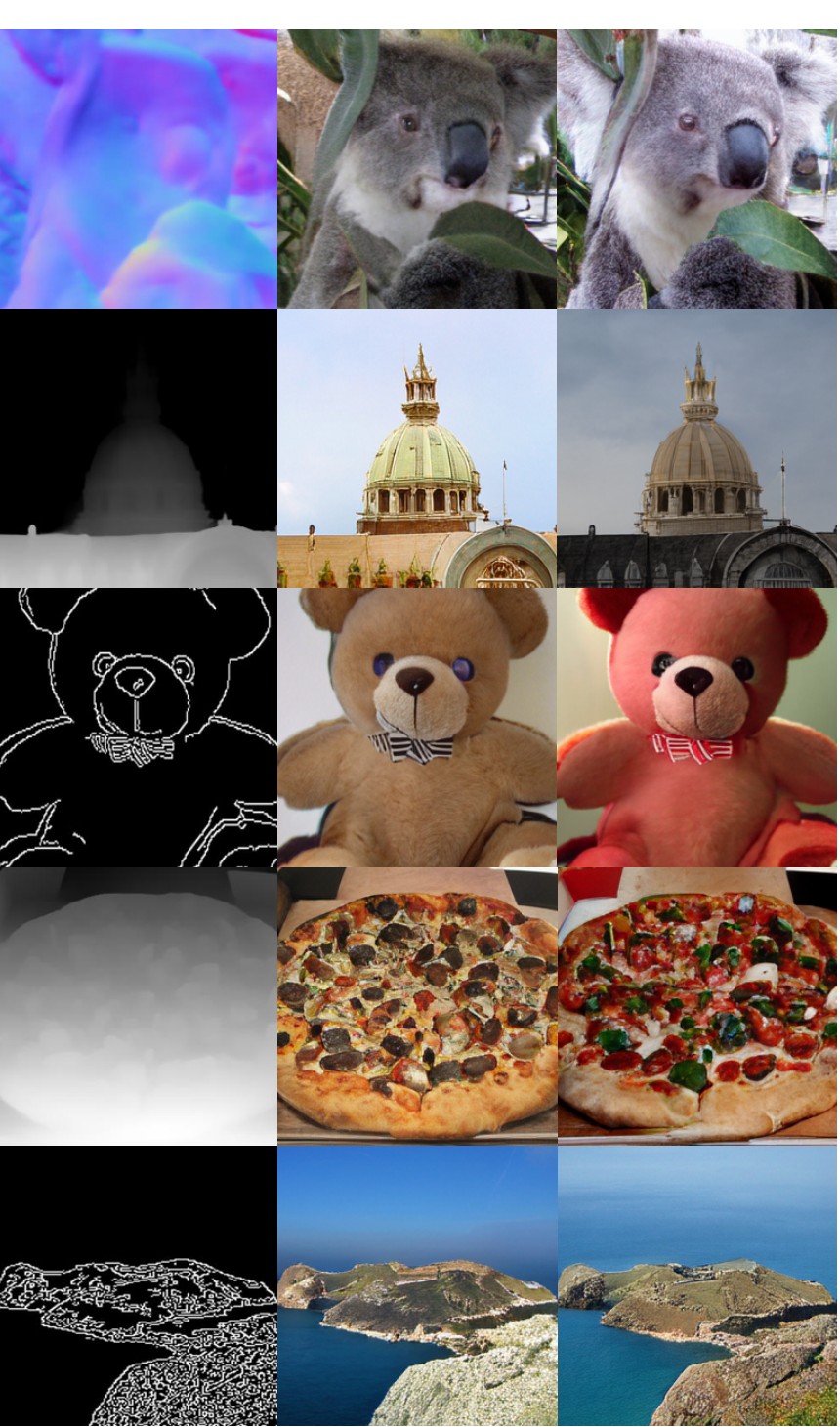

Figure 13: Visualization of ECM's conditional synthesis at 256×256 resolution.

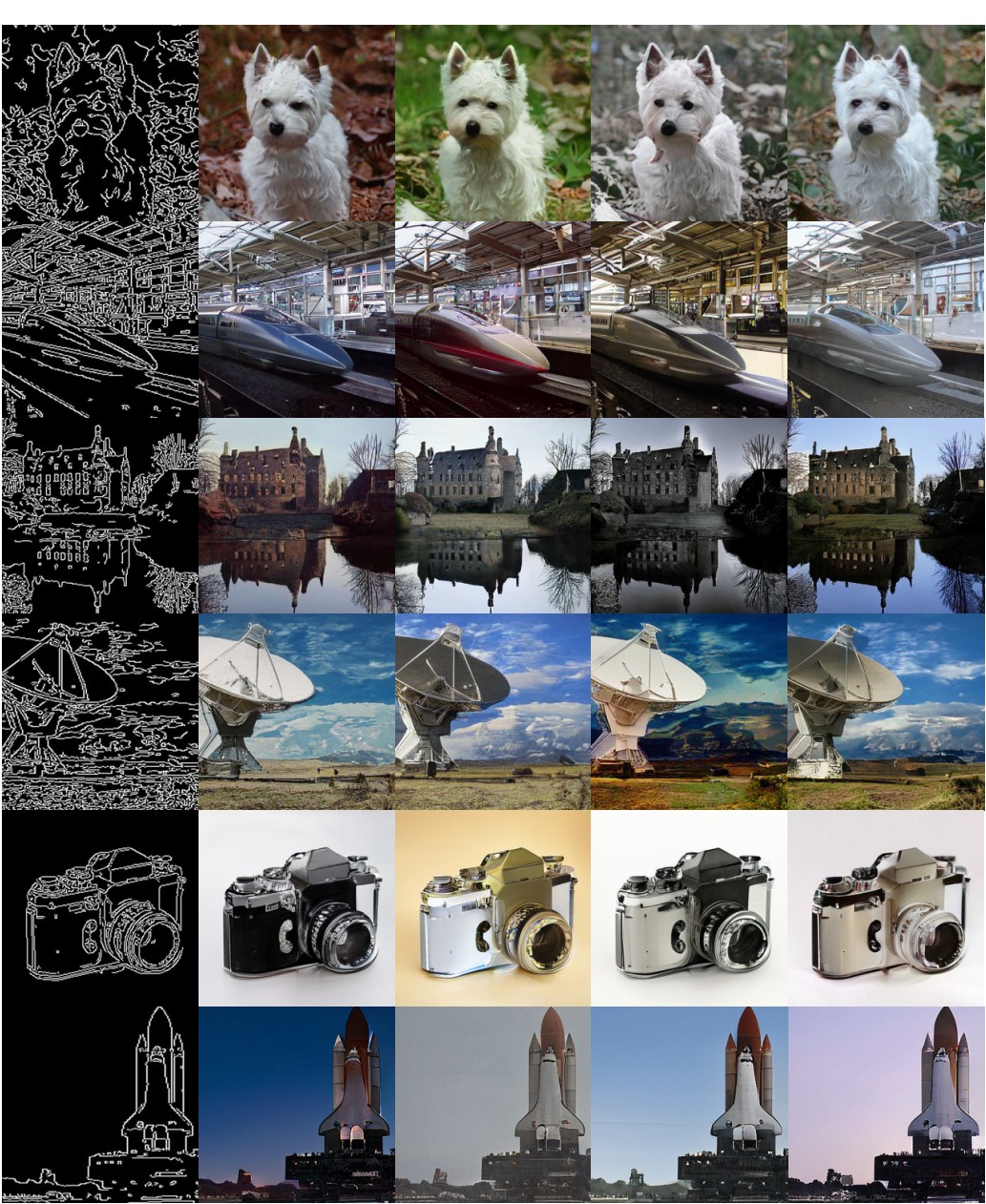

Figure 14: Visualization of ECM's conditional synthesis at 256×256 resolution.

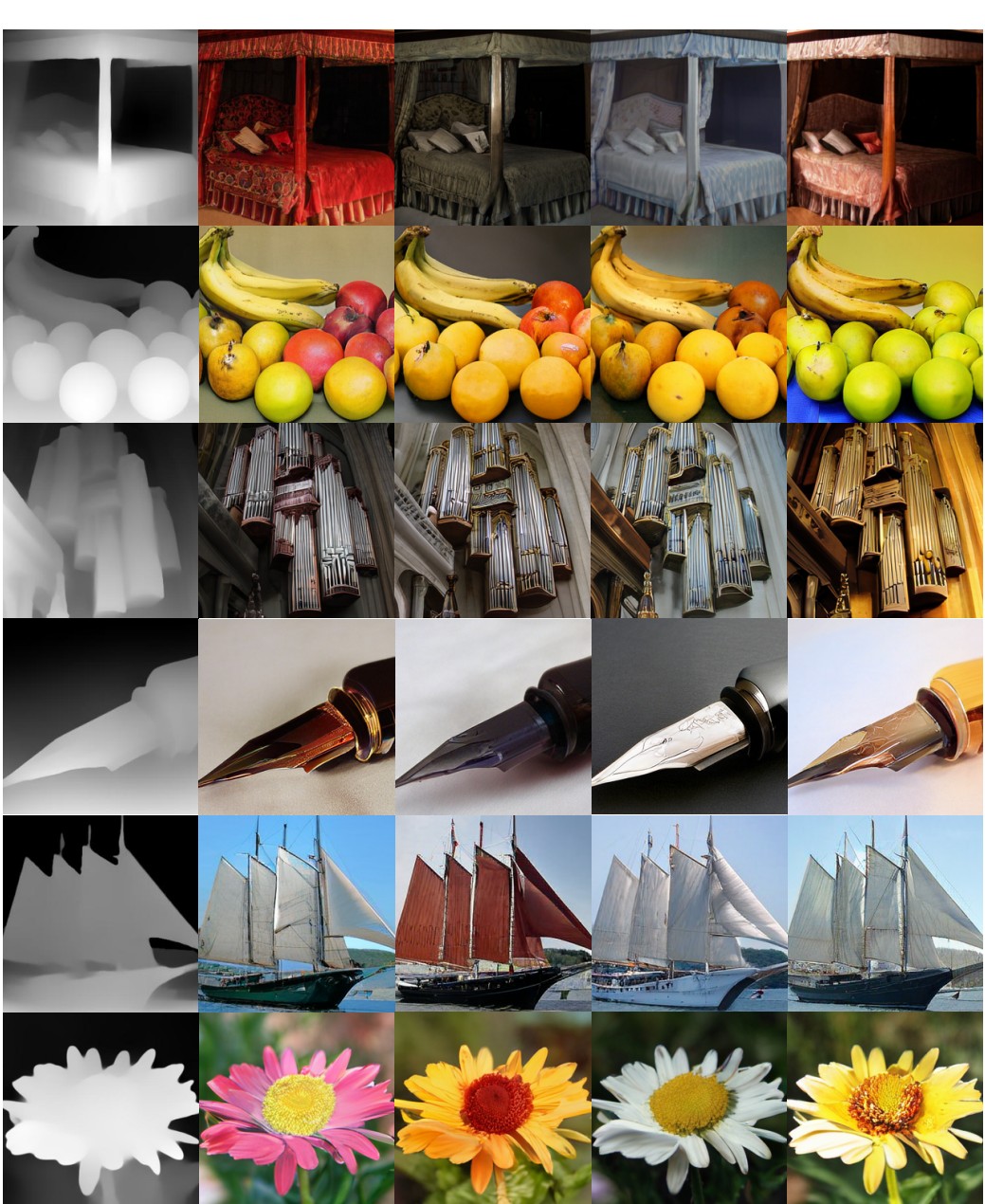

Figure 15: Visualization of ECM's conditional synthesis at 256×256 resolution.

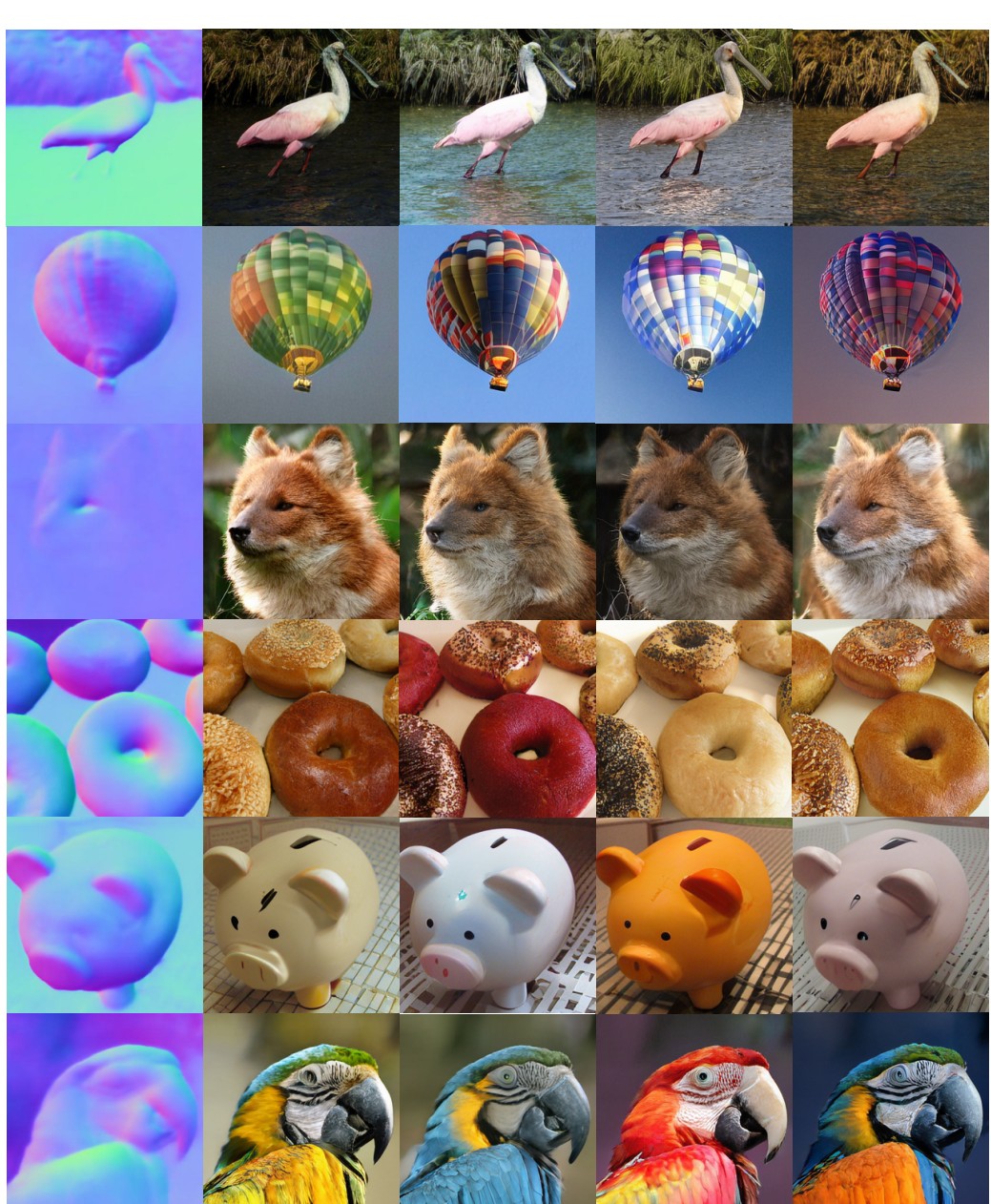

Figure 16: Visualization of ECM's conditional synthesis at 256×256 resolution.

