# OpenReview forum: "Efficient Conditional Generation on Scale-based Visual Autoregressive Models"
_ICLR.cc/2026/Conference — ICLR 2026 Conference Withdrawn Submission_

### Official Review · Reviewer_ruzd · 2025-10-31

**Soundness:** 2
**Presentation:** 2
**Contribution:** 1
**Rating:** 2
**Confidence:** 4

**Summary:**

Summary: This paper proposes the Efficient Control Model (ECM), a plug-and-play framework for adding spatial conditional control to pre-trained, scale-based autoregressive (AR) models like VAR. Instead of fine-tuning the base model or using a large, separate control network, ECM introduces a lightweight, distributed control module. This module consists of several small adapter blocks inserted throughout the frozen base model. The adapters feature context-aware attention layers and a shared, gated feed-forward network (FFN) to ensure efficient and coherent control. To further improve efficiency, the authors introduce an "early-centric sampling" strategy that prioritizes training on the initial, structurally critical generation scales, thus reducing the number of tokens processed. To compensate for the resulting under-training of later stages, a complementary temperature scheduling is applied during inference. Experiments show that ECM achieves high-fidelity conditional generation, outperforming existing baselines while being significantly more efficient in terms of trainable parameters and training time.

**Strengths:**

- This paper has a clear method presentation and some experiments.

**Weaknesses:**

- The method has zero novelty. It is just a simple modification of baseline.
- The experiments are limited.
- The model is mostly compared to ControlVAR while there are several more powerful model available recently. Like CAR, ControlAR. What about the comparison against them? Especially speed side.
- The paper presentation can be improved.

**Questions:**

NA

---

### Official Review · Reviewer_aeiv · 2025-11-01

**Soundness:** 3
**Presentation:** 3
**Contribution:** 2
**Rating:** 4
**Confidence:** 3

**Summary:**

The paper proposes Efficient Control Model (ECM), a lightweight conditional generation framework built upon frozen scale-based visual autoregressive (AR) models. Rather than fine-tuning the base model—which can introduce semantic conflict and high computational cost—ECM injects control via lightweight adapters. These include shared Feed-Forward Networks (FFNs) and individual attention modules, combined with an early-centric sampling strategy and temperature scheduling to improve training efficiency and output quality. Experiments across various conditioning modalities (Canny, depth, normals) show that ECM outperforms existing methods like ControlVAR and ControlAR, achieving better fidelity and diversity with lower training cost—even on smaller models.

**Strengths:**

- **No fine-tuning required**: ECM keeps the base model frozen, avoiding semantic conflict and distribution mismatch.
- **Low training cost**: Requires only 15 epochs; each epoch is 45% of ControlVAR’s training time.
- **Higher quality and diversity**: Outperforms prior methods (ControlVAR, ControlAR) on FID, IS, F1, and RMSE.
- **Early-centric sampling**: Focuses on early tokens to guide structure generation more efficiently.
- **Shared FFN with gated residuals**: Enables layer-wise collaboration without additional parameters.
- **Model scalability**: Performs robustly across 310M, 600M, and 1B parameter scales.

**Weaknesses:**

1.The proposed ECM framework is exclusively validated on scale-based AR models (i.e., VAR), with no empirical verification or theoretical analysis of its adaptability to non-scale-based AR architectures such as LlamaGen. This restriction confines the framework’s applicability to a narrow subset of AR models, undermining its broader impact on conditional generation research.
2.The verification experiments lack comprehensiveness. A majority of the paper’s validation focuses on class-conditional generation, while for the critical T2I task, experimentation is limited to VAR-CLIP alone—with only sparse data results provided in Appendix Tables 5 and 6. To better validate T2I effectiveness, comparisons with other state-of-the-art T2I scale-based AR models (e.g., HART, Infinity) are necessary. Furthermore, the paper only explores smaller VAR models (d16/d20), with results for VARd24 remaining cursory, hindering assessment of scalability.

**Questions:**

1. **Early-centric sampling tradeoff**: While it boosts efficiency, does it under-train later tokens? Are there adaptive or fine-grained alternatives?
2. **Temperature scheduling robustness**: The fixed (empirical) values, Thigh = 1.0, Tlow = 0.8, work well here, but how does this generalize to other tasks or resolutions?
3. **Shared FFN stability**: Multiple FFNs degrade performance. Was initialization a factor? Was pretrained FFN reuse attempted?

---

### Official Review · Reviewer_QJDo · 2025-11-03

**Soundness:** 3
**Presentation:** 3
**Contribution:** 3
**Rating:** 6
**Confidence:** 1

**Summary:**

The paper proposes the Efficient Control Model (ECM), a lightweight, plug-and-play framework that enables high-quality spatially conditioned image generation in scale-based autoregressive (AR) models without fine-tuning the base model. By using a distributed control architecture with shared feed-forward networks, early-centric training, and temperature scheduling, ECM achieves superior generation fidelity and diversity while significantly reducing both training and inference costs.

**Strengths:**

(1) It is a lightweight, plug-and-play framework that enables high-quality spatially conditioned image generation in scale-based autoregressive models without requiring fine-tuning of the pre-trained base model.
(2) The early-centric training strategy seems to be interesting and useful.
(3) It obtains better results than ControlVAR.

**Weaknesses:**

(1) The core contribution is not summarized.
(2) The paper highlights the disadvantages of existing control AR/VAR methods, such as ControlAR introduces computational overhea,d and CAR requires training lots of parameters. However, there are no comparisons between them to confirm that the method has solved the problem.
(3) Does the method work with vanilla AR generation models? I think the answer is yes. So, it requires experiments on vanilla AR models and directly compares with ControlAR.

**Questions:**

I would like to see the results of ECM with Vanilla AR models and apple-to-apple comparisons between ECM and ControlAR.

---

### Official Review · Reviewer_ed7R · 2025-11-04

**Soundness:** 3
**Presentation:** 3
**Contribution:** 3
**Rating:** 6
**Confidence:** 3

**Summary:**

The paper proposes ECM, a lightweight plug-and-play control module for conditional image generation using scale-based autoregressive models. ECM leverages adapter layers and an early-centric sampling strategy to improve efficiency and performance without fine-tuning the backbone. It achieves state-of-the-art results with fewer parameters and faster training.

**Strengths:**

-Novel plug-and-play design: ECM’s distributed adapter architecture (with partial FFN sharing and layer-specific gating) integrates condition signals without altering the base model. This design enables broad, adaptive control with minimal added parameters.
-Empirical performance: ECM outperforms existing AR-based methods in conditional synthesis quality and diversity. It achieves lower FID and higher Inception scores than baselines, indicating superior image fidelity under conditioning.
-Effective training strategy: The early-centric sampling greatly reduces the number of training tokens by prioritizing early-generation scales. This cuts computation and still yields strong results.

**Weaknesses:**

-Late-stage quality: Focusing on early tokens means the model is under-trained on final-generation details. The authors note that the generator is “less confident” at late stages, requiring a special temperature schedule. This workaround may not fully recover fine-detail fidelity.
-Limited evaluation scope: Experiments are restricted to 256×256 ImageNet with simple control maps (edges, depth, normals). It is unclear how ECM performs on higher resolutions, more complex conditions (e.g. semantic maps or text prompts), or real-world images beyond this setting.
-Comparisons and baselines: The paper compares mainly against other AR methods. It does not evaluate against diffusion-based conditional models or other parameter-efficient adaptation techniques (e.g. LoRA or prompt tuning) that might also achieve fast control. This leaves open how ECM stacks up against the broader set of conditional-generation approaches.

**Questions:**

-How sensitive is ECM to its hyperparameters (e.g. sampling strength, gating values, temperature schedule)? Is performance robust across choices or do certain settings degrade efficiency gains?
-Can the authors demonstrate ECM on additional tasks or datasets (e.g. higher-resolution images, semantic segmentation controls, or text-to-image conditioning) to test generality?
-Would combining ECM with other adaptation methods (like fine-tuning just the adapters or using LoRA) yield further improvements, or is ECM strictly intended as a zero-tuning solution?
-Are there failure modes or qualitative differences (e.g. in diversity or spatial accuracy) when using temperature scheduling versus uniform sampling? How stable are the results if scheduling is not applied?

---

### Note · Authors · 2025-11-12

**Comment:**

We appreciate the time and effort the reviewers dedicated to assessing our work. After careful consideration of the feedback, we have decided to withdraw our submission.

**Withdrawal Confirmation:**

I have read and agree with the venue's withdrawal policy on behalf of myself and my co-authors.